# Explaining Ionospheric Ion Upflow in the Subauroral Polarization Streams

Shuhan Li [ID], Jing Liu *[ID] and Qiaoling Li

Shandong Provincial Key Laboratory of Optical Astronomy and Solar-Terrestrial Environment, School of Space Science and Physics, Institute of Space Sciences, Shandong University, Weihai 264209, China
* Correspondence: liujing2019@sdu.edu.cn

**Abstract:** Ionospheric ion upflow is an important process for magnetosphere-ionosphere coupling via $O^+$ source for the magnetosphere. This process occurs frequently in the subauroral polarization stream (SAPS) region where the SAPS-enhanced ion-neutral frictional heating tends to push ions upward because of enhanced upward pressure gradient force. However, the SAPS-induced neutral wind transport by ion-neutral friction may also play an important role in triggering ion upflow, which has been rarely studied. In this work, the thermosphere-ionosphere-electrodynamics general circulation model (TIEGCM) with/without an empirical SAPS model has been employed to investigate the impacts of SAPS on ion upflow in the topside ionosphere. Our results separate different transport processes in the ion continuity equation, showing that SAPS can accelerate upward ambipolar diffusion along its channel because of ion-neutral frictional heating, but SAPS-induced horizontal neutral wind may have a comparable or even larger contribution to vertical ion drift when SAPS are fully developed. In addition, the neutral wind can induce both upward and downward ion drift in the SAPS region, depending on the direction of the neutral wind and the local geomagnetic declination and inclination.

**Keywords:** SAPS; ion upflow; neutral wind; TIEGCM; geomagnetic field





## 1. Introduction

The magnetospheric energy input at high latitudes can perturb the global thermosphere and ionosphere, causing thermospheric composition disturbance and propagation [1,2], driving a global wind surge from both polar regions [3], inducing enhanced and depleted ionospheric plasma densities [4,5] and forming subauroral polarization streams (SAPS) and storm enhanced density (SED) [6,7] ionospheric structures. SAPS, strong westward plasma flows driven by enhanced poleward electric fields, are frequently observed in the subauroral duskside ionosphere [8–10]. The location of SAPS remains inside the mid-latitude trough with low conductivity, independent of the variability of the location and the intensity of field-aligned currents (FACs) [11]. One possible mechanism of forming SAPS poleward electric field is that the electric fields in the magnetosphere, caused by the misalignment of the electron and ion boundaries at the plasmapause, map into the ionosphere along the magnetic field line, and the potential drop is mainly concentrated in the SAPS region with low conductivity [12]. Another view believes that an enhanced poleward electric field is required in the low conductive SAPS region to preserve current continuity between region-1 and region-2 FACs in the subauroral ionosphere [13]. SAPS thus should be an ionospheric response driven by the disturbed magnetosphere. Meanwhile, the severe friction between SAPS and the thermosphere can induce a further decrease in local conductivity [14,15] and be associated with ion upflow/outflow [16], modulating the FACs [17], to even feed back to the magnetosphere [18]. Therefore, SAPS and its ionospheric effects are important to our understanding of magnetosphere-ionosphere-thermosphere

coupling [19–23]. Numerical models, such as SAPS-TIEGCM [22], OpenGGCM-CTIM-RCM [24], LFM-TIEGCM-RCM [25], and SWMF [26], are applied to analyze the dynamics and electrodynamics related to SAPS.

Ionospheric ion upflow, an important role in the magnetosphere-ionosphere coupling, can often be observed at altitudes ranging from 200 to several thousand kilometers in the cusp and auroral region [27–29] or at the poleward boundary of the auroral oval [30,31], providing $O^+$ for the magnetosphere. Previous simulations and observations revealed that ion upflow could also occur in the mid-latitudes SAPS region, and the upflow velocities increased with the westward velocities of SAPS [32–36]. Statistical studies showed that there was a linear relationship between SAPS and ion upflow velocities [37]. SAPS-related ion-neutral frictional heating was essential for the formation of the upward ion flow in the dusk sector [16,37].

However, SAPS-induced vertical ion drifts are not homogeneous in the SAPS region. Previous simulations showed that SAPS could induce upward ion drift in the topside ionosphere or downward ion drift around 400 km [38,39]. SAPS-induced downward ion flow could also be observed in double-peak subauroral ion drift (DSAID) events [40]. More detailed studies are thus required to show the different influences of SAPS on vertical ion transport in the SAPS channel and their corresponding physical mechanisms. Therefore, we employed a thermosphere-ionosphere coupled model to analyze SAPS effects on vertical ion drifts. The following section describes the model and observations used in this work. The results are exhibited in Section 3. We discuss the explanations in Section 4.

## 2. Model and Observation Description

The NCAR TIEGCM is a first-principles three-dimensional thermosphere-ionosphere-electrodynamics general circulation model, solving the continuity, momentum, and energy equations self-consistently. TIEGCM is driven by the F10.7 solar index, solar irradiance [41], empirical high-latitude convection and precipitation obtained from the 3-h Kp index [42] or interplanetary magnetic field and solar wind [43], and lower atmospheric monthly climatology of tides specified by Global Scale Wave Model [44]. TIEGCM is used in this study with horizontal and vertical resolutions of 2.5° and 0.25 scale height ranging from ~97 to ~600 km.

SAPS effects were introduced by imposing an empirical SAPS velocity into subauroral ion drift at all altitudes. The empirical SAPS model was based on the statistical DMSP ion drift meter data [45], specifying SAPS distribution in magnetic latitude and local time by the 3-h Kp index [22]. At each time step in our simulation, the SAPS model was called for the grid point within 10° equatorward of the auroral precipitation boundary which was also specified by the Kp index. The empirical SAPS velocity was added to the ion velocity obtained from default-TIEGCM as the modification of SAPS. The ions with modulated velocity resulted in self-consistent ionospheric effects by dynamics and electrodynamics. In this study, the simulated results from default-TIEGCM and SAPS-TIEGCM were compared to analyze the effects of SAPS.

The solar wind parameters, interplanetary magnetic field, and geomagnetic activity index were obtained from the OMNI database. The ion drift velocities were observed by ion drift meters of DMSP satellites in situ at 840 km altitude. The observed horizontal cross-track ion drift velocity was projected to the zonal direction to compare with the simulated results.

## 3. Results

### 3.1. Observation and Simulation of 17–18 March 2015 Superstorm

A strong geomagnetic storm driven by coronal mass ejection and coronal hole high-speed streams occurred on 17 March 2015 [46], characterized by a sudden southward turning of interplanetary magnetic field (IMF) $B_z$ and an enhanced solar wind dynamic pressure. Figure 1 shows the solar geophysical conditions for this event on 17–18 March 2015. This event is the largest storm during the solar cycle 24, with a ~12 h and −18 nT

southward IMF Bz, the minimum SYM-H index (1-min resolution Dst index) −227 nT, and maximum 8− Kp with a duration of 12 h. During the main phase of the storm, multiple increases in the AE index indicated several obvious energetic particle injections which were related to SAPS occurrence [47].

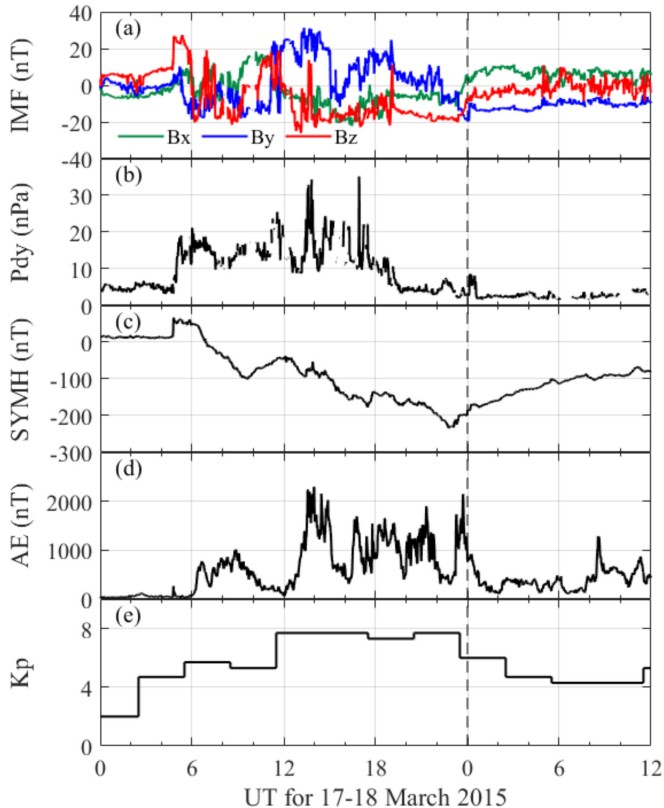

**Figure 1.** Solar geophysical conditions for 17–18 March 2015. (**a**) IMF $B_x$, $B_y$ and $B_z$ (nT) in GSM coordinates; (**b**) solar wind dynamic pressure $P_{dy}$ (nPa); (**c**) SYM-H index (nT); (**d**) AE index (nT); (**e**) Kp index.

Obvious SAPS/DSAID were observed by ground-based radars and DMSP space-craft during the main and early recovery phase during this event [16,26,48]. Figure 2 shows the eastward ion velocity ($V_{i\_E}$) observed by DMSP F15/F16/F17 and simulated by SAPS-TIEGCM on each satellite track at 2200 UT on 17 Mar 2015. For comparison with the simulations from SAPS-TIEGCM with a 2.5° × 2.5° horizontal resolution, the DMSP horizontal cross-track velocities were projected to the zonal direction and these velocities were smoothed in 42 s (~2.5°). In Figure 2a, the clockwise/anticlockwise red arrows are westward/eastward. The simulated $V_{i\_E}$ profile is similar to the observation. The simulation also reproduces the $V_{i\_E}$ westward two-peak structure in the dusk sector, in which one peak at lower latitude refers to the SAPS channel and the other peak at higher latitude refers to the duskside sunward convection. The magnitude of simulated SAPS maximum velocity over 1 km/s agrees well with the observations in magnitude at 2200 UT in Figure 2(b1–b3), while the latitudes of peaks shift ~8° owing to the underesti-mated size of the auroral oval for this great storm by empirical high-latitude Heelis model. Previous works have shown the effectiveness of SAPS-TIEGCM for subauroral dynamics and electrodynamics [22,49,50].

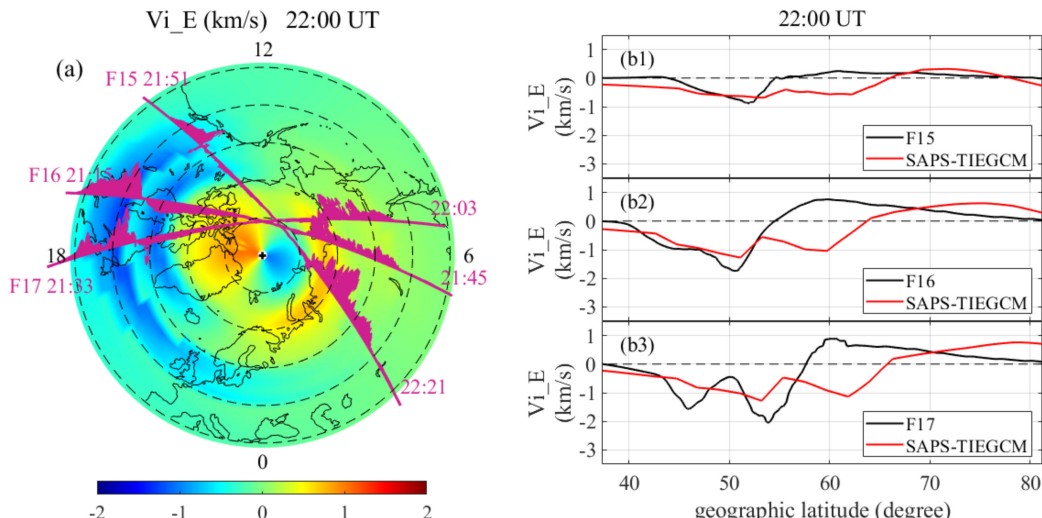

**Figure 2.** (**a**) Simulated eastward ion velocities $V_{i\_E}$ and the projection of DMSP F15/F16/F17 cross-track velocities in the eastward direction at 2200 UT on 17 Mar 2015; (**b1**–**b3**) simulated and observed $V_{i\_E}$ on different DMSP tracks.

During the main phase (2100–2300 UT) of this event, 50+ m/s enhanced upward ion drifts compared to a quiet day at a mean location in geodetic coordinates of (51.4°N, 86.7°W) and ~350 km were observed by Millstone Hill incoherent scatter radar (MHISR) [16]. Considering the observed high-speed westward flow and large ion temperature enhancements in the SAPS channel, Zhang et al. (2017) speculated the strong frictional heating due to SAPS drives significant atmospheric upwelling and corresponding ion upflow [16]. As shown in Figure 3, our simulation indicates that SAPS can induce the upward ion drifts with the same magnitude as MHISR observations.

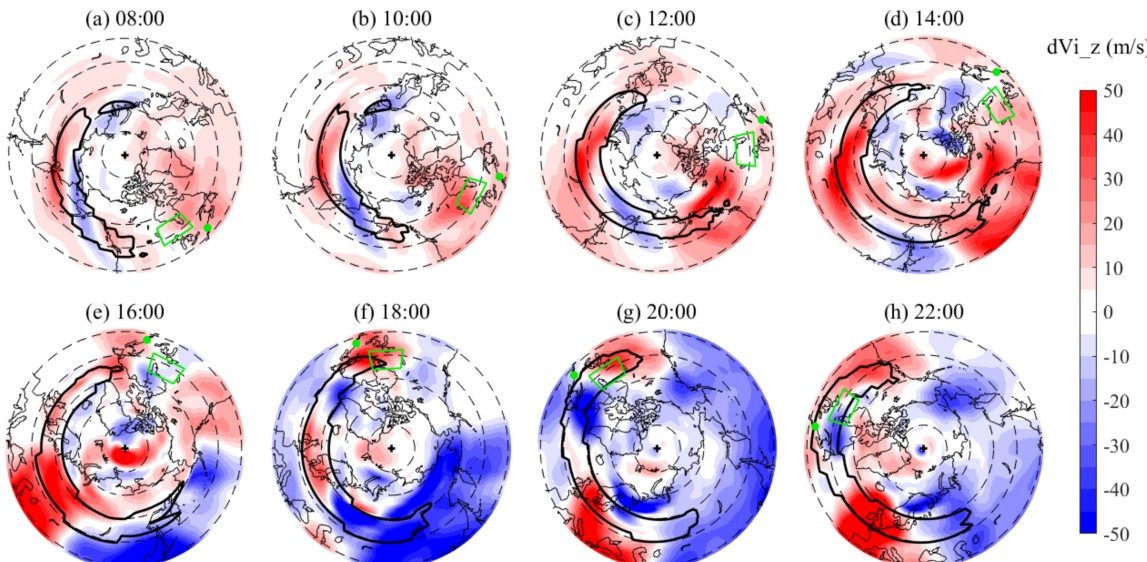

**Figure 3.** Simulated SAPS-induced upward ion velocities $dV_{i\_z}$ (the differences between the simulation results from TIEGCM with and without the SAPS model) at ~350 km at different UTs. The black contour lines are SAPS channels with $V_{i\_E} = -200$ m/s. The green points and boxes are the locations of MHISR and its approximate field of view, respectively.

Figure 3 shows the differences in simulated temporal evolutions of upward vertical ion drift velocities $dV_{i\_z}$ at ~350 km between SAPS-TIEGCM and default-TIEGCM, which

denote SAPS effects on ion drifts. The contour lines of the SAPS channel with westward velocities greater than 200 m/s are plotted as black solid lines at the dusk sector. From Figure 3a–h, modeled SAPS channel expands to more local times and shifts to a lower latitude as compared to the observations reported by He et al. [20]. The zonal extension and equatorward movement of SAPS are caused by the enhanced SAPS velocity and the extended high-latitude convection, respectively.

Figure 3 indicates that SAPS-induced vertical ion drifts depend on longitude and UT. During 1200–1600 UT, Kp reached the maximum value of 8− (Figure 1e), and the modeled SAPS also were maximized. Meanwhile, obvious ion upflow appeared in almost the whole SAPS channel. Subsequently (1800–2200 UT), downward ion flows occurred at dusk in the SAPS channel, while upward ion drifts remained in the afternoon and evening. The green points and boxes in Figure 3 refer to the location of MHISR (42.6°N, 71.5°W) and its field-of-view approximation (51.4 ± 4°N, 86.7 ± 10°W) [16]. Figure 3g,h indicates that MHISR enters a SAPS-induced upflow region with upward velocities of ~50 m/s during 1800–2000 UT. The magnitude of simulated upward ion drifts is consistent with Millstone Hill observations [16]. Therefore, we use the model to analyze the contributions of different physical processes in the SAPS region to the ion upflow.

### 3.2. Physical Mechanisms of SAPS-Induced Ion Upflow

Intense westward plasma transport and ion-neutral friction can affect vertical ion transport around the SAPS channel [16,22,51]. The upflow-related physical processes, which are difficult to separate observationally, can be simulated to determine their respective contributions.

In TIEGCM, ion $O^+$ transport is divided into three terms, including transport induced by electric fields, neutral winds, and ambipolar diffusion. Therefore, the vertical ion velocity $V_{i\_z}$ can be described by

$$V_{i\_z} = V_{i\_E \times B\_z} + V_{i\_w\_z} + V_{i\_d\_z}, \tag{1}$$

where $V_{i\_E \times B\_z}$, $V_{i\_w\_z}$, and $V_{i\_d\_z}$ are upward vertical ion velocities caused by electric fields, neutral winds, and ambipolar diffusion. $V_{i\_w\_z}$ and $V_{i\_d\_z}$ can be derived as

$$V_{i\_w\_z} = -(V_{w\_E} \sin(D) + V_{w\_N} \cos(D)) \cos(I) \sin(I) + V_{w\_z} \tag{2}$$

$$V_{i\_d\_z} = -\frac{k \sin^2 I}{m_i \nu_{in}} \left( \frac{T_i + T_e}{N_i} \frac{dN_i}{dh} + \frac{d(T_i + T_e)}{dh} + \frac{m_i g}{k} \right), \tag{3}$$

where $V_{w\_E}$, $V_{w\_N}$, and $V_{w\_z}$ are eastward, northward, and upward neutral winds; $D$ and $I$ are geomagnetic declination and inclination; $k$, $m_i$, $\nu_{in}$, $T_i$, $T_e$, $N_i$, and $g$ are Boltzman constant, $O^+$ mass, collision frequency between $O^+$ and neutrals, $O^+$ temperature, electron temperature, $O^+$ density, and gravitational acceleration, respectively [52–54]. It is noted that the distributions of SAPS-induced ion upflow velocities in its channel are different in Figure 3e,h. The vertical ion drifts are almost upward in the whole flow channel at 1600 UT (Figure 3e) and are upward in the afternoon and downward ion drifts at dusk at 2200 UT. The SAPS-induced ion $O^+$ vertical velocity distributions at 1600 and 2200 UT and the contributions of separated transport processes are shown in Figure 4.

Figure 4 indicates that both SAPS-induced neutral wind and ambipolar diffusion contribute to the variation of vertical ion velocity, and the SAPS-induced wind transport breaks the homogeneous distributions of ion upflow in the longitudinal direction. Ambipolar diffusion dominates at 1600 UT, driven upward ion velocities in almost the whole SAPS region, while wind transport turns to dominate at 2200 UT after SAPS attain their maximum intensities for 10 h (from 1200 UT to 2200 UT). Previous studies showed that the frictional heating in the SAPS region produced by intense horizontal ion-neutral relative motion was speculated to be associated with strong ion upwelling/upflow, leading to the uplifted ionosphere or upward plasma ambipolar diffusion [16,37]. And SAPS-induced ion upflow was formed by frictional heating and consequent ionosphere lifting [16,37]. During solar flare events, the enhanced upward pressure gradient may drive plasma upward ambipolar

diffusions and thus an ion upflow [54]. However, our results in Figure 4 have shown that SAPS-induced strong wind effects on vertical ion drift have equal or more importance to ion-neutral frictional heating effects. At 1600 UT, the maximum ion upward velocity induced by ambipolar diffusion can reach ~30 m/s in Figure 4d, while wind effects on upward ion drifts are ~20 m/s in Figure 4c. Moreover, Figure 4c also shows that SAPS-related wind can cause vertical ion transport in polar cap. At 1800 UT, comparing Figure 4e,g, wind effect can even dominate the pattern of ion vertical drift. It means that the influences of SAPS on neutral winds can gradually accumulate and may eventually dominate the ion upflow.

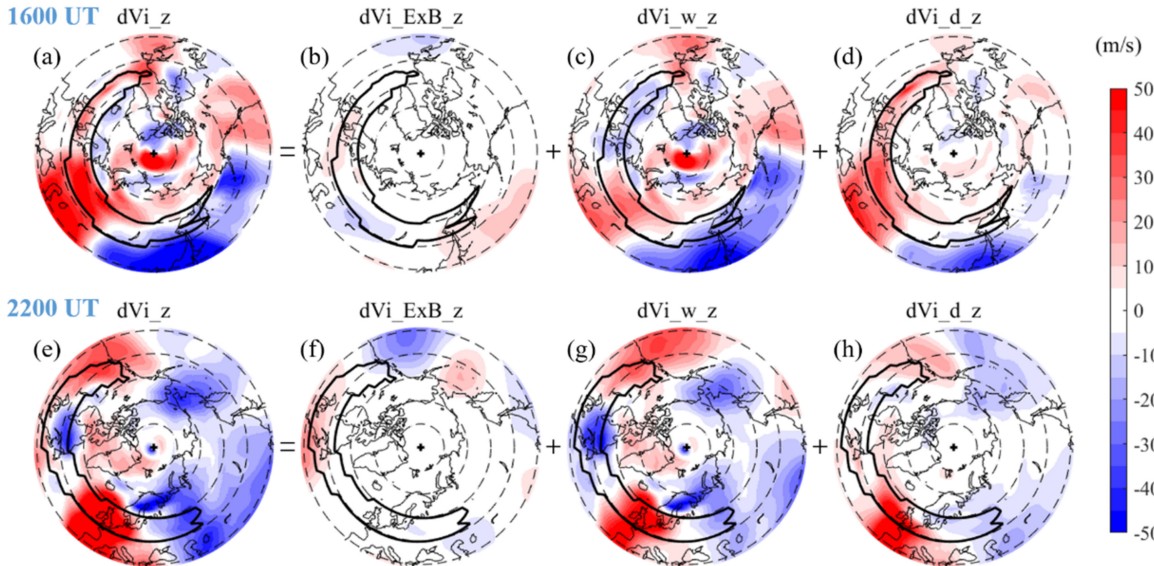

**Figure 4.** SAPS-induced vertical ion velocities due to (**a**,**e**) the combined effects of electric fields, neutral wind and ambipolar diffusion, (**b**,**f**) only electric fields, (**c**,**g**) neutral winds, and (**d**,**h**) ambipolar diffusion at 1600 (**upper panels**) and 2200 UT (**bottom panels**).

To determine how SAPS modulate wind effects on ion upflow, considering that SAPS mainly cause the enhancement of westward wind because of large ion-neutral velocity difference and frequent collision between them. The equatorward wind is also very efficient in pushing ions upward in the presence of inclined magnetic field lines, Figure 5 shows the effects of SAPS-induced winds in different directions on upward ion drifts (Figure 4g). SAPS-enhanced westward wind tends to push ions downward at dusk and upward at noon in Figure 5a. In the afternoon/dusk sector, the upward/downward ion flow caused by the SAPS-enhanced equatorward/poleward wind has the same magnitude of velocity as the westward wind. An enhanced equatorward wind in the evening sector leads to local ion upflow. Moreover, disturbed global meridional wind can result in a wider range of vertical ion transport even far from the SAPS channel. The weaker contribution of vertical wind indicates that SAPS-induced horizontal winds play a major role in pushing plasmas upward.

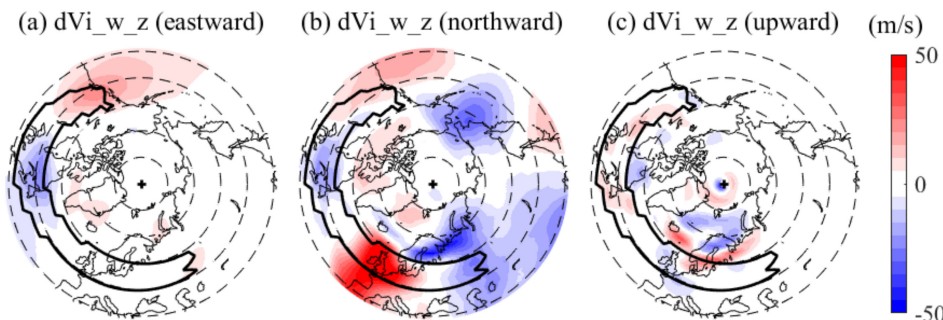

**Figure 5.** Vertical ion velocities caused by SAPS-induced (**a**) eastward wind, (**b**) northward wind, and (**c**) upward wind at 2200 UT.

## 4. Discussion

SAPS can change the horizontal wind gradually by ion-neutral collision. Figure 6a,b show the eastward and northward differential neutral winds due to SAPS effects at 2200 UT. The simulated SAPS-enhanced westward wind is ~300 m/s and the poleward wind in eastern North America is ~100 m/s, which is consistent with the Millstone Hill observations and GITM simulations [55]. The poleward wind in the SAPS channel is mainly driven by the Coriolis force, forming a clockwise vortex-like wind at the north edge of the SAPS channel [55]. Considering the geomagnetic inclination is nearly invariant ~72° (Figure 6d), the meridional winds pushing ions to drift along the field line depend on the wind direction (Figure 6b), thus the vertical ion velocities as Figure 5b. As SAPS-induced zonal wind in its channel is nearly westward (Figure 6a), the geomagnetic declination determines the direction of ion vertical movement pushed by the strong westward wind, which is upward when $D > 0$ and downward when $D < 0$, forming the pattern in Figure 5a.

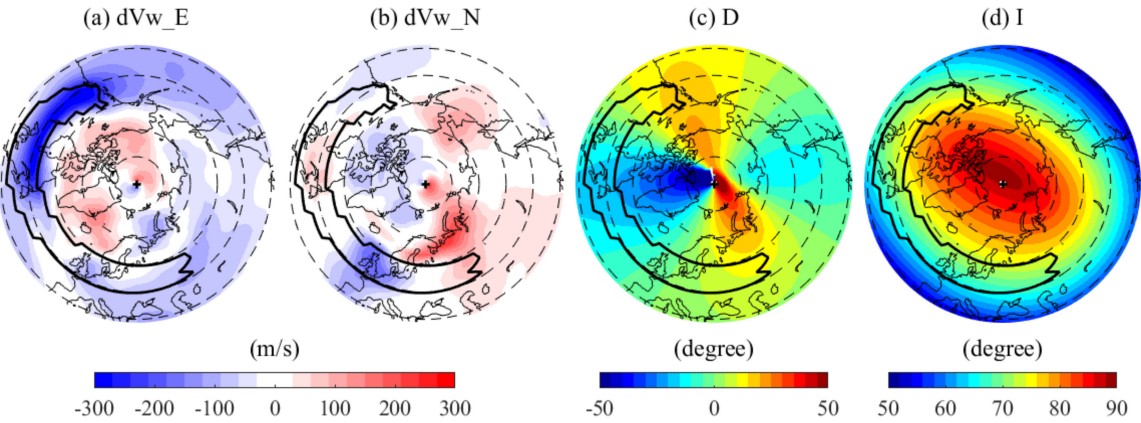

**Figure 6.** The difference of (**a**) eastward neutral wind and (**b**) northward neutral wind between TIEGCM with and without SAPS. The distribution of (**c**) geomagnetic declination and (**d**) inclination.

Figure 7 shows the temporal evolution of differential (a) eastward ion velocities $V_{i\_E}$, (b) eastward ($V_{w\_E}$) and northward ($V_{w\_N}$) neutral wind velocities, (c) vertical ion velocities induced by different components of neutral wind, and (d) geomagnetic declination and inclination between SAPS-TIEGCM and default-TIEGCM at 1800 LT in the SAPS channel. SAPS velocity reached its maximum at 1330 UT (Figure 7a), while the peak of SAPS-enhanced westward wind appeared at 1450 UT (Figure 7b). SAPS-induced changes in the neutral wind have a delay of several hours with respect to the development of SAPS [47,55]. From 1330 UT to 2400 UT, more stable SAPS lead to a persistent poleward wind at dusk, which might be formed by the Coriolis force [50,55]. Figure 7b–d indicates that SAPS-induced westward wind can move ion upward or downward depending on local geomagnetic declination. When $D > 0$ from 1100 UT to 1750 UT, westward wind may

drag ions upward as shown in Figure 8a, while SAPS induces downward ion drift when local $D < 0$. However, the vertical ion drift caused by SAPS-induced meridional wind depends on wind direction as the local geomagnetic inclination is approximately 75°. SAPS-induced poleward wind may lead to a downward plasma drift, while the equatorward wind has the opposite effects as shown in Figure 8b.

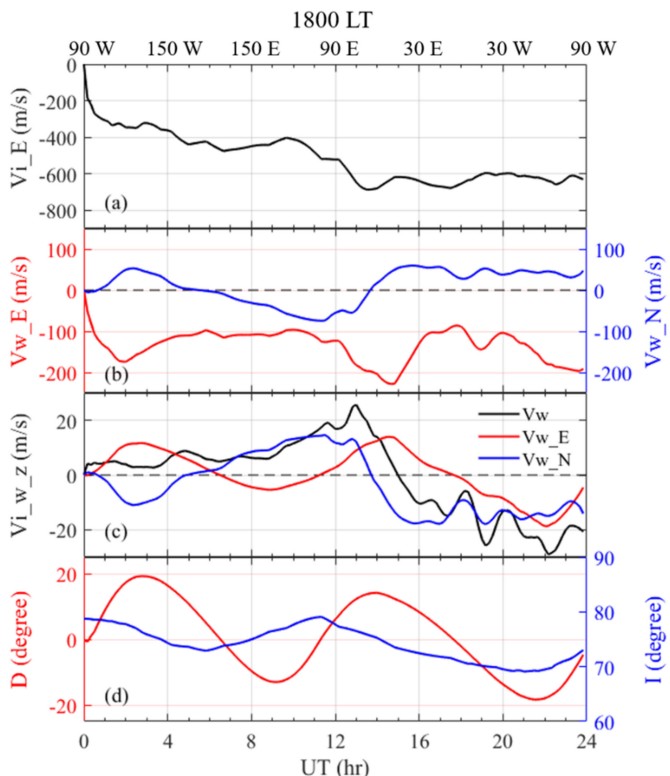

**Figure 7.** The averaged differential (**a**) eastward ion velocities $V_{i\_E}$, (**b**) eastward ($V_{w\_E}$) and northward ($V_{w\_N}$) neutral wind velocities, (**c**) vertical ion velocities induced by different components of neutral wind, and (**d**) geomagnetic declination and inclination between SAPS-TIEGCM and default-TIEGCM in SAPS channel at 1800 LT.

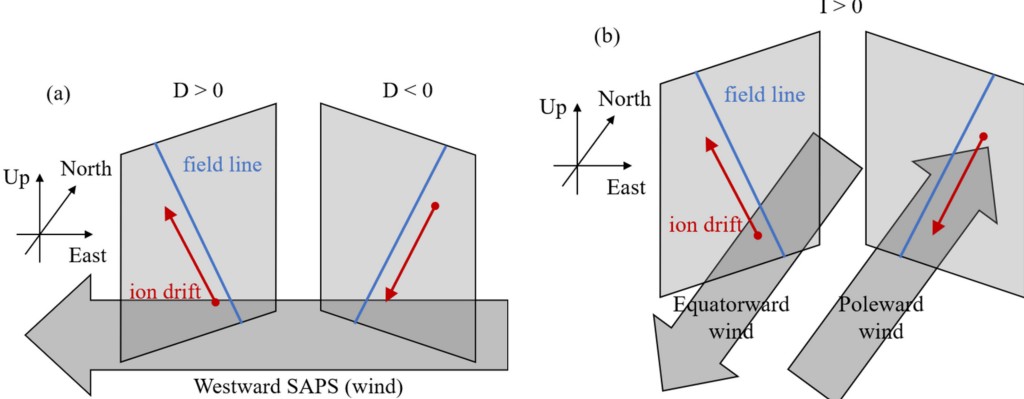

**Figure 8.** A schematic illustration of vertical ion drift in the Northern Hemisphere caused by (**a**) westward wind in the presence of declined magnetic field lines, and (**b**) meridional wind in the presence of inclined magnetic field lines in the SAPS channel.

More generally, SAPS-induced ionospheric ion upflow associated with neutral wind depends on wind velocity and the location of concerned. Zou et al. [56] reported that the velocity of SAPS-related westward wind was about 21% of the velocity of SAPS

flow during the non-storm time, while Wang et al. [57] showed that the statistical SAPS-related westward wind had a velocity of about 35% of the plasma velocity during active time. Considering that the statistical mean velocities of SAPS for moderate magnetic activity (Kp < 4) and active period (Kp > 4) were respectively 600–800 m/s and 800–1000 m/s depending on season [58], SAPS-induced westward wind velocity should be ~200–300 m/s, in agreement with our simulation in Figure 7b. In the subauroral region, the maximum upward drift velocity related to geomagnetic declination is ~7% of westward wind in Figure 7. It indicates that SAPS-enhanced westward wind can provide ~15–20 m/s maximum ionospheric ion upward drift at 350 km. Meanwhile, SAPS-induced meridional wind can have a similar magnitude contribution to ion upflow as zonal wind does. SAPS thus can induce ~30–40 m/s maximum velocity of ion upflow in the F-region, which is nearly 4% of SAPS velocity. However, SAPS velocities can exceed 2000 m/s occasionally, leading to over 80 m/s ions upflow velocities induced by neutral wind. In these cases, combined effects of neutral wind and ion-neutral frictional heating thus can induce more than 150 m/s ions upward drifts in the SAPS region as observed [16], providing plasma sources for subsequent outflow processes at higher altitudes.

## 5. Conclusions

In this work, the impacts of SAPS on vertical ion drift in the Northern Hemisphere have been studied, which shows the importance of SAPS-induced vertical ion drifts at ~350 km due to neutral wind effects. The contribution of SAPS-induced wind transport by ion-neutral collision to vertical ion drift can be comparable or greater than SAPS-enhanced ambipolar diffusion by frictional heating as neutral wind disturbances build up. The disturbed wind can drive ions upward or downward in the SAPS channel, depending on the wind direction and geomagnetic declination and inclination. In the Northern Hemisphere, SAPS-enhanced westward wind can drive ion upward/downward flow at the locations where D > 0/D < 0, and SAPS-induced equatorward/poleward wind may move ion upward/downward. The maximum upward ion velocity induced by SAPS-related neutral wind can reach ~30–40 m/s on average, which is ~4% of SAPS velocity.

**Author Contributions:** Writing—original draft, S.L.; Writing—review & editing, J.L. and Q.L. All authors have read and agreed to the published version of the manuscript.

**Funding:** This work is supported by the National Natural Science Foundation of China 42122031, 42074188, 42104147; Shandong Provincial Natural Science Foundation ZR2022JQ18, the Strategic Priority Research Program of Chinese Academy of Sciences Grant No. XDB 41000000; and Guangdong Basic and Applied Basic Research Foundation 2020A1515110242.

**Acknowledgments:** We acknowledge the use of data from the Chinese Meridian Project. The solar wind parameters are obtained from the OMNI 2 database (https://spdf.gsfc.nasa.gov/pub/data/omni/high_res_omni/, accessed on 14 June 2022). The DMSP data can be obtained from the Madrigal database (http://cedar.openmadrigal.org/, accessed on 17 August 2022).

**Conflicts of Interest:** The authors declare no conflict of interest.

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
