# Peer review of "Explaining Ionospheric Ion Upflow in the Subauroral Polarization Streams"

_remotesensing, doi:10.3390/rs14246315_

Round 1

Reviewer 1 Report

The authors analyzed ionospheric ion upflow in the SAPS based on TIEGCM. The simulation results are compared and evaluated with DMSP observations. This work has great soundness and potential value for understanding the magnetosphere-ionosphere coupling. I think this manuscript could be accepted with minor revision.

1.     Line 146, the sentence mentioned that upward velocity of 50m/s in the simulation. Please clarify if this value is determined by experience or other prior information.

2.     Line 186-188, SAPS-induced strong wind effects on vertical ion drift have equal or more 187 importance to ion-neutral frictional heating effects. Please explain this conclusion more with the figure.

Reviewer 2 Report

  This paper provides a detailed study of the ionospheric ion upflow impacted by SAPS during a strong geomagnetic storm by TIEGCM. First the TIEGCM results are validated by good data-model comparison of this study and other  previous studies. Then the author divide the vertical ion drift velocity into three parts, vertical component of E cross B drift, vertical neutral winds and vertical ion velocity due to ambipolar diffusion. Through diagnostic analysis, they found that SAPS can accelerate upward ambipolar diffusion along its channel due to ion-neutral frictional heating. However, SAPS-induced horizontal neutral winds have a comparable or even larger contribution to vertical ion drift when SAPS are fully developed. Furthermore,  the neutral wind can induce both upward  and downward ion drift in the SAPS region, which is related to the direction of the neutral wind and the  local geomagnetic declination and inclination. The paper is well organized and written. I recommend to have a minor revision before it is suitable for publication

In the introduction, the author shall first give a brief description of storms impact on both thermosphere and ionosphere. It shall include the thermospheric composition disturbance generation and propagation, positive and negative ionospheric plasma density variation, SAPs and Storm enhanced density (SED). Then the author can introduce the SAPs in detail

For the reference of thermosphere composition disturbance, the author can cite

Cai, X., Burns, A. G., Wang, W., Qian, L., Solomon, S. C., Eastes, R. W., et al. (2021). Investigation of a neutral “tongue” observed by GOLD during the geomagnetic storm on May 11, 2019. Journal of Geophysical Research: Space Physics, 126, e2020JA028817. https://doi.org/10.1029/2020JA028817

And other related papers

For reference of positive and negative ionosphere plasma density variation, the typical references are

Yue, X., W. Wang, J. Lei, A. Burns,Y. Zhang, W. Wan, L. Liu, L. Hu, B. Zhao, and W. S. Schreiner (2016), Long-lasting negative ionospheric storm effects in low and middle latitudes during the recovery phase of the 17 March 2013 geomagnetic storm, J. Geophys. Res. Space Physics, 121, 9234–9249,doi:10.1002/2016JA022984.

Rajesh, P. K., Lin, C. H., Lin, C. Y.,Chen, C. H., Liu, J. Y., Matsuo, T., et al. (2021). Extreme positive ionosphere storm triggered by a minor magnetic storm in deep solar minimum revealed by FORMOSAT-7/COSMIC-2 and GNSS observations. Journal of Geophysical Research: Space Physics, 126, e2020JA028261. https://doi.org/10.1029/2020JA028261

For reference of SED, the author can cite

Foster, J. C., Coster, A. J., Erickson, P. J., Holt, J. M., Lind, F. D., Eideout, W., et al. (2005). Multiradar observations of the polar tongue of ionization. Journal of Geophysical Research, 110, A09S31. https://doi.org/10.1029/2004JA010928

Liu, J.,W.Wang, A. Burns, X. Yue, S. Zhang, Y. Zhang, and C. Huang (2016), Profiles

of ionospheric storm-enhanced density during the 17 March 2015 great storm, J. Geophys. Res. Space Physics, 121, 727–744, doi:10.1002/2015JA021832.

 In the model description in section 2, the author missed several aspects. They shall point out that Solar EUV irradiance input is based on Solomon and Qian (2005). Also they shall give the source of ionization due to auroral precipitation.

Line 66-67 here the author shall also clarify that the lower boundary is based on monthly climatology of tide

Line 91 12h of what?? Duration time or the main phase time??

Figure 1  AE cannot be less than 0, so you can just set the y axis starting from 0.

Figure 2, so the upward red arrows stand for eastward direction??
